# Rapid Multianalyte Microfluidic Homogeneous Immunoassay on Electrokinetically Driven Beads

**DOI:** 10.3390/bios10120212

**Published:** 2020-12-21

**Authors:** Pierre-Emmanuel Thiriet, Danashi Medagoda, Gloria Porro, Carlotta Guiducci

**Affiliations:** Laboratory of Life Sciences Electronics, École Polytechnique Fédérale de Lausanne, 1015 Lausanne, Switzerland; danashi.medagoda@epfl.ch (D.M.); gloria.porro@epfl.ch (G.P.); carlotta.guiducci@epfl.ch (C.G.)

**Keywords:** bead-based immunoassays, dielectrophoresis (DEP), three-dimensional microelectrodes, on-chip incubation, acute kidney injury diagnosis, multimarker analysis, microfluidic-based diagnostics

## Abstract

The simplicity of homogeneous immunoassays makes them suitable for diagnostics of acute conditions. Indeed, the absence of washing steps reduces the binding reaction duration and favors a rapid and compact device, a critical asset for patients experiencing life-threatening diseases. In order to maximize analytical performance, standard systems employed in clinical laboratories rely largely on the use of high surface-to-volume ratio suspended moieties, such as microbeads, which provide at the same time a fast and efficient collection of analytes from the sample and controlled aggregation of collected material for improved readout. Here, we introduce an integrated microfluidic system that can perform analyte detection on antibody-decorated beads and their accumulation in confined regions within 15 min. We employed the system to the concomitant analysis of clinical concentrations of Neutrophil Gelatinase-Associated Lipocalin (NGAL) and Cystatin C in serum, two acute kidney injury (AKI) biomarkers. To this end, high-aspect-ratio, three-dimensional electrodes were integrated within a microfluidic channel to impart a controlled trajectory to antibody-decorated microbeads through the application of dielectrophoretic (DEP) forces. Beads were efficiently retained against the fluid flow of reagents, granting an efficient on-chip analyte-to-bead binding. Electrokinetic forces specific to the beads’ size were generated in the same channel, leading differently decorated beads to different readout regions of the chip. Therefore, this microfluidic multianalyte immunoassay was demonstrated as a powerful tool for the rapid detection of acute life-threatening conditions.

## 1. Introduction

Acute kidney injury (AKI) is a life-threatening condition characterized by a rapid loss of kidney function [1]. In developed countries, AKI occurs in 20% of hospitalized adult patients and 25% of pediatric patients receiving intensive care [2], and its diagnosis is critical to improve survival. One of the consequences of AKI is the disruption of homeostasis, inducing an accumulation of waste products normally removed by the kidneys, which can lead to severe damages throughout the body. If treated quickly, the effects of AKI are reversible, notably through fluid resuscitation and medication [3] but they can lead to death of the patient without proper intervention.

Currently, AKI is diagnosed through monitoring of the patient’s urine output volume and measurement of the level of serum creatinine in blood [3]. Creatinine quantification suffers from diverse limitations, namely, interferences with drugs such as antiretroviral drugs [4], variations in basal creatinine levels between patients, and most importantly a long delay (36 to 48 h) between the occurrence of AKI and a detectable increase in serum creatinine concentration [5]. This delays the diagnostics, with possibly critical consequences. Numerous potential alternative AKI biomarkers are currently investigated by research groups worldwide [6]. Here we focused on the most promising ones, Cystatin C and Neutrophil Gelatinase-Associated Lipocalin (NGAL). Cystatin C is a molecule present in all tissues and filtered by the kidneys. Its concentration spikes in serum 24 h following injury [7,8]. NGAL is a protein that can be found in neutrophils and some epithelia including renal tubules. As AKI damages the kidneys’ epithelium, such disorder induces an increase of NGAL concentration in serum within four hours [9,10]. A combined analysis of both NGAL and Cystatin C would allow for early diagnosis of AKI, reducing the risk of false negatives [9].

The critical requirements of timeliness in the diagnosis of AKI call for fast analytical devices to perform the analysis of the relevant biomarkers directly within intensive care units or emergency facilities. The detection of such markers in clinical settings relies on immunoassays carried out by means of bulky analyzers, such as Abbot Architect i1000SR and c4000, used, respectively, for the quantification of NGAL and Cystatin C. Commercial point-of-care (PoC) systems for the individual detection of these targets are also available on the market, namely, the Triage^®^ system of Alere Inc. for the detection of NGAL through a lateral flow assay [11] and the Cube^®^ of Eurolyser for Cystatin C levels’ quantification. We aimed to concomitantly detect Cystatin C and NGAL using an immunoassay approach that could be engineered into a PoC device. Such multiple analysis would be an effective mean to obtain an accurate diagnosis of AKI in the first day following the injury. In order to minimize the number of assay steps, our device performed both the binding of the analyte molecules from the sample and the readout phase. We implemented such detection on antibodies-decorated beads to perform sandwich immunoassays. Indeed, functionalized microbeads suspended in a microfluidic channel provided a fast and effective collection of analytes in-flow. However, the gathering of beads into large clusters was required to obtain a high readout fluorescent signal. The spatial handling of beads is commonly performed either through magnetic forces [12] or by means of mechanical restrictions [13]. These approaches suffer from limitations hindering their use in portable devices, namely, a difficult integration into a portable platform for magnetic devices and clogging issues for mechanical systems. Electrokinetics manipulation of beads through the generation of gradients of electrical fields appears as a potential solution for beads’ manipulation in a compact and highly integrated system.

Dielectrophoresis (DEP) is a physical phenomenon appearing when placing a polarizable particle into a non-uniform electric field. It has been successfully employed to sort particles according to their size, shape, and dielectric properties [14,15]. It also proved to be valuable in the diagnostics field through enrichment of targeted protein [16], stream focusing in flow cytometry [17], and spatial confinement of decorated beads [18,19,20]. Iswardy et al. [18] implemented a DEP-based biosensing platform for the diagnosis of Dengue virus. The analyte is captured on beads functionalized with antibodies specific to one of the virus proteins and is held in place thanks to DEP forces. Ramon-Azcon et al. [19] developed a device detecting pesticide residues in wine. Decorated beads are maintained against the flow with DEP and exposed to the sample of interest. A similar technique was proposed by Park et al. [20], combining the manipulation of beads with DEP forces with a polarization-based preconcentration approach to increase the analyte concentration in the vicinity of beads. However, this solution imposes limitations on the flow rate and, thus, on the amount of analyte accessible for detection. Most importantly, for all the abovementioned technologies, the levels of DEP forces achieved in buffers of regular conductivity are not sufficient to efficiently manipulate the beads; therefore, they require the use of highly diluted solutions (low ionic force) [21], which limits their overall sensing performance.

Here, we introduce high-aspect-ratio vertical electrodes in the microfluidic channel to enable highly effective manipulation of microbeads by electrokinetics in minimally diluted buffers (5x). In our system, microbeads decorated with specific antibodies were incubated with analytes on-chip along DEP trapping regions in presence of flow, drastically reducing mass transport issues. They were later moved and concentrated to detection regions, where the optical fluorescence signal of the markers was amplified by clustering multiple beads. Our device allowed carrying out both analyte binding and signal readout under the continuous flow of the same reagents, avoiding rinsing steps.

In this paper, we present the first microfluidic multianalyte platform for rapid detection of AKI. The simultaneous assessment of Cystatin C and NGAL levels we achieved allows for rapid and accurate diagnosis of AKI within a large temporal window after the injury and with a matching performance compared with a state of the art ELISA kit.

## 2. Materials and Methods

### 2.1. Chip Microfabrication

The microfluidic system consists of a 4-mm-wide main channel, spanned by six rows of vertical electrodes. Two inlets allow for sequential injection of beads and reagents, subsequently disposed through a single outlet. The height of both the microfluidic channel and the vertical electrodes is 50 µm.

The platform is fabricated through an additive process on a glass substrate [22]. A detailed illustration of the process flow is reported in Appendix A. After sputtering of a Ti/Pt/Ti (20/200/20 nm) layer on the entire wafer surface (Pfeiffer Spider 600, Pfeiffer Vacuum, Asslar, Germany), planar metal lines are patterned with photolithography and ion beam etching (Veeco Nexus IBE 350, Veeco, Plainview, TX, USA). Successively, in order to insulate the metal lines from the liquid, a 300-nm layer of oxide is sputtered on the wafer. This layer is opened through dry etching (SPTS Advanced Plasma System (APS), SPTS technologies, Newport, UK) in the electrically active regions of the device. Cylindrical vertical pillars in SU-8 photoresist (Microchem 3025, Microresist Technologies, Berlin, Germany) are then deposited on the exposed metal pads connected to the oxide-passivated lines. A layer of Ti/Pt (20/200 nm) is sputtered onto the entire wafer surface. This layer is subsequently removed through vertical ion beam etching, leaving the metal only on the vertical pillars’ sidewalls. Once the fabrication of the electrodes has been carried out, the SU-8 microfluidic channel is patterned at the same height as the electrodes. Finally, the device is sealed with a polydimethylsiloxane (PDMS) coverslip, bonded to the chip by a 3-aminopropyl triethoxysilane (APTES) treatment and a baking step at 150 °C for 2 h. This results in the integration of electrically active 3-D structures within the microfluidic channel, shown in Figure 1.

### 2.2. DEP-Based Manipulation of Beads

Dielectrophoresis (DEP) is a phenomenon that occurs to a polarizable particle placed in a non-uniform electric field. The presence of an electric field triggers the formation of a dipole in a polarizable particle. The dipole experiences a net force if the electric field is spatially non-uniform, which leads the particle to move. The direction of the force induced by DEP on the particle depends on the relative polarizability of the particle and of the surrounding medium. If the particle is more polarizable than the surrounding medium, the particle dipole will be oriented along the electric field, while a particle less polarizable than the surrounding medium results in a dipole oriented against the electric field. The force experienced by a spheroid particle with a radius R placed in an electric field E can be written as:(1)$$F_{DEP}=2\pi R^{3}\varepsilon_{medium}Re\left(f_{CM}\left(\omega \right)\right)\nabla E^{2}$$
where $\varepsilon_{medium}$ is the dielectric permittivity of the medium and $f_{CM}\left(\omega \right)$ is the Clausius-Mossotti factor, which is a function of the permittivity and, consequently, of the polarizability of the particle and the surrounding medium [23].

Furthermore, a spherical particle injected into the microfluidic channel is dragged by the flow according to the following equation:(2)$$F_{drag}=6\pi \eta RV_{0}$$
where *η* is the fluid viscosity, *R* the sphere radius, and *V*_0_ the fluid constant velocity at infinity.

In our device, the electric field is generated by linearly arranged vertical electrodes. If the line of electrodes is normal to the flow, the beads flowing through the channel experience a DEP force directed against the drag force (Figure 2a). Provided that the DEP force is sufficient to compensate for the drag force, the beads are immobilized upstream in the vicinity of the electrodes’ line as illustrated in Figure 2c. The beads can be kept at this position while experiencing a continuous flow of analyte and reagents, thus allowing their binding to the beads’ surface. However, if the row of electrodes is placed at a specific angle with respect to the flow, the DEP force will not be directed against the drag force and we will observe a net force heading the bead in a specific direction (Figure 2b and Appendix A and Appendix A). This “surfing” phenomenon, illustrated in Figure 2d, allows for the displacement of beads along diagonal lines and for their accumulation at a dedicated location in the channel, which results in an increase of the readout signal and improves the device sensitivity [24].

Equation (1) establishes the dependence between the size of the beads and the DEP force they will experience. Each line of electrodes can generate a different DEP force and can consequently trap and displace specific beads depending on their size. In this way, beads of different sizes can be driven to distinct regions on the chip (Figure 2f). If beads of different sizes are functionalized with different antibodies, our approach allows a multianalyte analysis with a single fluorescent channel.

### 2.3. Microfluidic Analytical Device Operation and Experimental Setup

The microfluidic chip for DEP-based immunoassays is shown in Figure 3a. It consists of a single channel with two inlets and a single outlet featuring linear arrangements of three-dimensional electrodes, patterned to obtain incubation lines (horizontal) and concentration lines (diagonal).

Bead-based sandwich immunoassays were performed on our platform as illustrated in Figure 3b,c. The fluorescently labeled detection antibodies (dAbs) were spiked in the sample prior to injection in the chip. Before undergoing injections of reagents solutions, the chip was primed with buffer solution (fetal bovine serum, FBS, diluted five times in deionized water) in order to prevent unspecific bindings to the channel walls. (1) After priming, beads functionalized with capture antibodies (cAbs) were injected in the device and held in suspended small clusters (few tens of beads) upstream of the incubation lines, as illustrated in step 1 in Figure 3c. The number of captured beads was controlled through visual inspection. During this phase, the microbeads–cAbs solution, 10 µL 0.5% (*w/v*) in 1 mL of diluted FBS, was delivered from one inlet at 2 µL/min, maximal flow that could be applied without having beads escaping the incubation region, while the incubation line exerted a holding force (20 Vpp, 1 MHz). The obtained small beads’ aggregates were spatially confined by the DEP force, while kept in slight agitation by the flow: This turbulent motion favors the convective transport of target molecules. (2) After a sufficient number of beads was collected (1–2 ms), the beads–cAbs solution flow was stopped and the solution of antigen–dAbs complexes was immediately dispensed from the other inlet at 2 µL/min. The antigen–dAbs solution was flushed in the chip for 15 min. During this incubation step on-chip, the binding between the analyte–dAbs and the cAbs led to the formation of the complete sandwich assay on the beads’ surface. (3) After incubation, the beads were released by turning off the electrical signal. Concomitantly, the concentration line was activated (20 Vpp, 1 MHz) and the beads were led to “surf” in-flow along the diagonal concentration lines until they reached a region where they were trapped against the microfluidic wall, as shown in Appendix A. (4) The beads carrying the sandwich assays were, hence, accumulated to enhance the total fluorescence signal. We noticed that the so-obtained clusters, imaged in CY5 fluorescence channel at 5000 ms exposure, presented the same intensity per µm^2^ provided that their footprint was equal or larger than 50 µm^2^. (5) Finally, the beads were released from this concentration region by deactivation of the electrodes’ line.

A similar protocol can be applied to run two distinct immunoassays in parallel on the same channel. Two different beads’ populations (2-µm and 6-µm diameter) were functionalized with cAbs for Cystatin C and NGAL, respectively. The beads–cAbs solution was obtained by adding 10 µL 0.5% (*w/v*) 2-µm beads and 10 µL 0.5% (*w/v*) 6-µm beads in 1 mL 5-fold diluted FBS. The incubation step was performed on two different lines (the line upstream was activated by applying a 14-Vpp, 1-MHz signal and the line downstream by applying a 20-Vpp, 1-MHz signal) in presence of a 0.4 µL/min flow, maximal flow that was applied while holding the small 2-µm beads against the flow. The line upstream was set to exert a weaker electric field so that beads of larger size were trapped by the first line, while the smaller ones could pass through, to be successively trapped by the second line, with a trapping efficiency of 90% and 70% for the 6-µm and the 2-µm beads, respectively. After beads’ clustering, incubation was performed flushing a solution containing the analyte–dAbs complexes of both Cystatin C and NGAL, with the dAbs labeled with CY3 and CY5 fluorophores, respectively. Following incubation, the two beads’ populations carrying the two distinct sandwich assays were released and concentrated at different locations on-chip. To do so, the two accumulation lines were activated (line upstream electrical stimulus: 14 Vpp, 1 MHz; line downstream electrical stimulus: 20 Vpp, 1 MHz), and the flow was set to 0.8 µL/min, maximal flow allowing accumulation of 2-µm beads. The two distinct clusters were imaged in CY3 and CY5 fluorescent channels with 5000-ms exposure times. On average 20 experiments (incubation, accumulation, and beads’ release) could be performed before the appearance of clogging issues preventing the chip from further use.

The experimental setup employed for on-chip experiments included two PHD ULTRA™ syringe pumps (Harvard Apparatus, Holliston, MA, USA), a TG5012A function generator (Aim-TTi, Huntingdon, UK), an optical fluorescence microscope (Leica DM2500M) with Y3 and Y5 filter cubes (Leica, Wetzlar, Germany), and a camera (ORCA-Flash4.0LT, (Hamamatsu Photonics, Hamamatsu, Japan). The electric signal was provided to the chip through a homemade printed circuit board (PCB), depicted in Appendix A.

### 2.4. Antibodies’ Conjugation to Beads and to Labeling Fluorescent Molecules

Polystyrene beads covalently coated in Streptavidin (Spherotec Inc.) were purchased at sizes of 2-µm diameter (binding capacity = 0.42 nmol/mg) and 6-µm diameter (binding capacity = 0.14 nmol/mg). The 2-µm beads were incubated with biotinylated Cystatin C monoclonal capture antibodies (Cyst13-biotinylated, Novus Biologicals) and the 6 µm-beads were incubated with biotinylated NGAL polyclonal capture antibodies (Human Lipocalin-2/NGAL-biotinylated antibody BAF1757, R&D Systems, UK) for at least four hours. They were then resuspended and incubated for 1–2 h in 1% bovine serum albumin (BSA, Sigma Aldrich, St. Louis, MO, USA) in 0.1 M phosphate buffer (PB, Sigma Aldrich, St. Louis, MO, USA) for blocking. After blocking, beads were washed four times through a procedure of centrifugation, supernatant removal, and resuspension in a solution of 0.05% Tween-20 (Millipore, Burlington, MA, USA) in 0.1 M PB. After the final washing steps, beads were stored in 0.1 M PB at 4 °C. Both antigens for Cystatin C (Human Recombinant Cystatin C, Novus Biologicals, UK) and NGAL (Human Lipocalin-2/NGAL, CF, R&D Systems, UK) were acquired and used as is. Monoclonal detection antibodies for Cystatin C (Cyst24-Dylight 550, Novus Biologicals, UK) were purchased with a Dylight 550 fluorophore, while monoclonal detection antibodies for NGAL (Human Lipocalin-2/NGAL Antibody MAB17571R, R&D Systems, UK) were labeled with a fluorophore using a Lightning-Link Rapid Alexa Fluor 647 antibody labeling kit (Expedeon, San Diego, CA, USA). Prior to experiments, antigen and detection antibodies were incubated together for 15 min in fetal bovine serum (FBS, Sigma Aldrich, St. Louis, MO, USA) diluted five times in MilliQ water.

### 2.5. Data Analysis

All images and videos acquired in the scope of this paper were analyzed using ImageJ software (Fiji). For each cluster, the neighboring background signal was calculated and subtracted from the mean intensity in the regions of interest, to account for possible variations of the background intensity. With this approach, we quantified the normalized fluorescence signals displayed in Figure 4, Figure 5 and Figure 6 and Appendix A. In addition to this, in order to plot the calibration curves of Figure 5, Appendix A, the average signal of the negative control was subtracted from each measurement point. This allowed us to remove artefact effects induced by unspecific binding of the labeled antibody with the beads.

## 3. Results and Discussion

### 3.1. Immunoassay Incubation On-Chip

The on-chip incubation of antibody-functionalized beads with antigens and detection antibodies took place on dedicated horizontal incubation lines. We maintained beads decorated with cAbs against the flow and concentrated them in small clusters of about 10 to 20 beads. Significant variability was observed in the size of the growing clusters, reflecting inhomogeneities in the flow lines within the microfluidic channel. Such inhomogeneities were mainly caused by microfabrication defects or debris accumulation at the entrance of the chip. Then, a solution containing the dAb–NGAL complexes was injected at a flow rate of 2 µL/min for 15 min. The binding of the dAb–NGAL complexes to the cAb-decorated beads was observed and quantified in real time by fluorescence measurements of the CY5 channel (Figure 4 and Appendix A). Figure 4a illustrates the dependence of measured fluorescence intensity of the small clusters on the concentration of NGAL in the injected solution. Such behavior could be quantified with a minute-range resolution, as shown in Figure 4b. A steady increase of fluorescence signal could be recorded after 15 min upon injection of NGAL–dAbs complexes at concentrations in the range of 0.5–100 ng/mL. Interestingly, none of the plots were reaching a plateau, suggesting the persistence of a transient binding regime. Different concentrations corresponded to similar fluorescent intensities, e.g., 5 ng/mL and 10 ng/mL. Thus, they might have appeared perilous to resolve due to a significant standard deviation. This variability in the clusters’ fluorescent signal was caused by the varying size of the small clusters formed near the incubation lines. Nonetheless, the purpose of the horizontal incubation lines was to allow for incubation to happen in the most favorable conditions and not to enhance the readout of the fluorescence intensity. This step took place in the accumulation regions, presented in the next section.

We designed the DEP holding action at the horizontal incubation lines in a way that the size of the clusters would be maintained sufficiently small to keep the accumulated beads in slight agitation. This approach allows for uniform binding of analytes in-flow and access to the whole surface offered by the beads. In fact, in the case of large clusters, the reagents would be depleted at the downstream portion of the aggregate, an issue that can be noticed in previous works [18]. This limitation inspired us for the creation of two separate lines for incubation and accumulation of beads, presented in Figure 3.

Moreover, previous approaches employing DEP force to immobilize beads and expose them to reagents relied on the use of planar electrodes to generate electric fields [20]. This approach relies on a simple fabrication process but leads to the creation of high electrokinetic forces only in close proximity to the chip’s surface that features the electrodes. Consequently, as we moved toward the opposite side of the channel, the DEP force experienced by a bead decreased. Vertical electrodes employed in our device, instead, generated a homogeneous electrical field over the entire channel height, ensuring that all the beads entering the incubation region experienced the same DEP force. The height of the channel was then no longer limited and could be substantially increased, to 50 µm in this device and possibly larger, as only the microfabrication process dictates the extension of the microchannel height. Increasing the height of the channel helped us increase the reagent flow rate in the microfluidic chamber by a factor of 10 or more compared to similar detection platforms [18,20], improving the collection of analytes that may be only present at low concentration in serum.

Another advantage deriving from the three-dimensional electrodes employed to generate DEP forces is the use of minimally diluted solutions. Currently, as DEP forces are weakened in high ionic strength solutions [25], most DEP-based microfluidic platforms are forced to operate in extremely diluted (low ionic force) solutions, thus drastically limiting the actual detection capabilities of their systems due to the consequent dilution of the analytes’ concentration [20]. Indeed, diluting the sample of interest by a large factor will consequently reduce the output signal. In comparison, our device successfully performed beads’ collection and analyte binding in an only 5× diluted serum, a dilution factor commonly found in commercial biomarker assays [26]. A slight dilution has proven to be even suitable for biomarker analysis, as it reduces matrix effect while maintaining favorable binding conditions [27].

Furthermore, the integration of the incubation step on-chip appears as a key milestone in the process of embedding our technology in a point-of-care device (PoC). In fact, it would limit the variance and errors in the concentration readout that might derive from additional manipulations [28].

### 3.2. Immunoassay Performance

The step directly following on-chip incubation is the accumulation of beads in dedicated areas, to obtain a larger signal and, thus, increase the sensitivity of detection. This section aimed to assess the impact of carrying out beads’ incubation and aggregation in our microfluidic system, comparing our approach both to a bulk process of incubation of the reagents and to the detection of the signal from single beads. To do so, the experimental plan described in Figure 5b was designed and implemented. Beads decorated with NGAL cAbs were incubated either on-chip on the horizontal *incubation lines* with the antigen–dAbs complex, as described in the previous section, or off-chip, by placing the Eppendorf containing the beads and the complexes in a rotating mixer. The obtained beads were then either clustered on-chip, as shown in Figure 5a, or observed sparsely on a microscope slide, and the corresponding fluorescent signal was acquired and plotted, as in Figure 5c.

The comparison between plots 1 and 2 of Figure 5d shows the impact of measuring the fluorescence signal of the microbeads in the accumulation regions on-chip vs. measuring the fluorescence of single beads on plates. In fact, in both cases, the prior incubation of beads with the analyte and dAb was performed identically off-chip in vials. We observed a signal increase of 3.5-fold in the case of beads concentrated in specific locations on the chip, demonstrating the validity of our approach to enhance the signal by locally increasing beads’ density to get a larger signal. Such amplification allows the detection of 1 ng/mL NGAL concentration in 5-fold diluted FBS, while this concentration could not be resolved by observing the fluorescent intensity of single beads. Common approaches for beads’ accumulation involve the application of magnetic forces [29,30] or DEP forces [18]. The increase in signal that we found to be a consequence of the beads’ accumulation is in line with what has been previously reported in literature [29].

On the other hand, the influence of incubation conditions on the attained signal could be analyzed comparing plots 2 and 3 of Figure 5d. In this case, the clustering of beads was carried out in the microfluidic platform in both experiments, while the incubation was performed either on-chip or off-chip. On-chip incubated beads reached the same level of antigen binding as the off-chip incubated beads. This demonstrates that the on-chip process matches the performance of a standard incubation in turbulent regime.

The integration of the incubation step within the microfluidic system is a key step in the design of autonomous lab-on-chip platforms. Recent approaches emphasize the need to maintain a certain level of agitation for beads during incubation, in order to maximize the interaction between beads and target analytes in the solution [12,31,32]. Indeed, a local increase of the convection phenomenon in close vicinity of the beads would reduce mass transport issues and speed up the supply of analyte. The incubation step we implemented on the horizontal lines appears to be as efficient as a standard turbulent off-chip incubation, even though it was previously reported that turbulent incubation performs better than exposure to continuous flows [33,34]. Our solution thus succeeded in providing an incubation as efficient as the gold standard off-chip methodology.

The NGAL concentrations that could be detected with our device ranged from 1 to 100 ng/mL, with a limit of detection of 1 ng/mL, calculated using a three-times standard deviation approach. However, as the concentrations expressed in the calibration curve refer to the five times diluted serum, this interval can be translated into a 5–500-ng/mL detection range in non-diluted serum. This interval covers the clinical NGAL concentration values observed in healthy patients (around 80 ng/mL) and patients suffering AKI (above 300 ng/mL) [10,35]. Furthermore, the standard deviation calculated with our platform appears to be small enough to distinguish healthy patients from ill patients. Indeed, the clinical procedure for AKI diagnosis, defined as an increase of more than 100% of the NGAL basal concentration [10,36], is resolvable with our platform, which makes our device suitable for the rapid diagnostics of kidney injury. In order to compare the performance of our platform with a state-of-the-art method, the same samples were tested by ELISA (Appendix A), exhibiting similar performance, with comparable sensitivity over a 1–100-ng/mL concentration range. Notably, the assay time could be shortened down to 15 min with our platform, versus 4 h required to run the ELISA test. Table 1 describes the advantages of our approach over existing solutions for the detection of NGAL.

All experiments carried out within this publication were run in 5 times diluted serum sample (FBS: fetal bovine serum). As none of the previous studies employing DEP to perform immunoassay on beads relies on serum [18,20], we are the first to implement the detection of analytes with DEP in a representative medium.

### 3.3. Simultaneous On-Chip Analysis of AKI Biomarkers NGAL and Cystatin C

This section aimed to investigate the possibility to concomitantly quantify the concentration of two biomarkers on a single chip. The two chosen analytes, NGAL and Cystatin C, spike into serum at different stages of the kidney injury [9]. Three hypothetical scenarios were defined based on clinical data: a “healthy” patient who presents basal levels for both NGAL (75 ng/mL) and Cystatin C (250 ng/mL), an “early-stage” patient presenting a spike in NGAL (300 ng/mL) and basal Cystatin C levels (250 ng/mL), and finally a “late-stage” patient presenting a spike in Cystatin C (1000 ng/mL) and basal NGAL levels (75 ng/mL). Two antibodies selective to NGAL or Cystatin C were conjugated to 6-µm and 2-µm beads, respectively. Sets of beads with different sizes are necessary to separate and localize beads in different areas on the device (See Materials and Methods section).

In order to describe the efficiency of beads’ separation, NGAL and Cystatin C detection antibodies (see Figure 3b) were labeled with fluorophores emitting in different spectral regions, respectively in CY5 and CY3 regions. The spectral overlap between fluorophores was also investigated to ensure negligible spectral overlap due to the fluorophore emission ranges (See Appendix A).

Figure 6a shows the achieved spatial separation of beads based on their size, with the 6-µm and the 2-µm beads, respectively, appearing as in red or green. The performance of this multianalyte approach is quantified in Figure 6b for the three patients described above. As the fluorophores emitting in CY3 and CY5 regions were different in brightness, the comparison between the absolute fluorescent signals was not relevant.

For the CY5 channel (Figure 6b, red), corresponding to the fluorescence of 6-µm beads capturing NGAL collected in cluster 1, the detection of the NGAL spike in the “early” case scenario appears to be clearly resolved. The presence of some signal in cluster 2, shown in Appendix A, for all scenarios suggests that some 6-µm beads could cross the first electrical barrier and get trapped at the downstream location. This contamination, evaluated at 20% of the signal obtained in the 6-µm beads’ clusters, was due to defects in the electrical contact between some of the vertical pillars and the planar electrodes, leading to the leaking of beads through the lines and unwanted gathering in the downstream region, which could be reduced by an optimization of the microfabrication process. The signal produced by NGAL beads appears lower than the one observed with a similar concentration in the previous section, which was due to the smaller flow rate used in this experiment for incubation, 0.4 µL/min vs. 2 µL/min.

Regarding the CY3 channel (Figure 6b, green), corresponding to the fluorescence of 2-µm beads capturing Cystatin C accumulated downstream in cluster 2, the detection of the Cystatin C spike in the “late” case scenario can be resolved vs. normal conditions. The corresponding calibration curve for Cystatin C on-chip detection can be found in Appendix A. We observed a limit of detection of 0.5 ng/mL and a detection range covering concentrations from 0.5 to 200 ng/mL, which suits the clinical requirement for detection of AKI in patients [37]. Furthermore, the resolution at 50 ng/mL was calculated and estimated to be of 11 ng/mL. Contaminations can also be noticed (Appendix A) in cluster 1 for all scenarios. We estimated that contaminations were equivalent to 25% of the signal obtained in the 2-µm beads’ clusters and arose from a tendency of small beads to stick together and with large beads, therefore, forming clusters while incubating on horizontal lines. Those clustered beads then behaved as larger beads and, thus, accumulated in cluster 1. Such effect could be mitigated through the introduction of a surfactant in the reaction solution. Moreover, assuming that the assay will be calibrated to result in comparable fluorescence signals corresponding to physiological basal levels of NGAL and Cystatin C (healthy patient), a 20% variability due to contaminations would not prevent us from detecting a 100% increase in one of the marker concentrations, as an effect of an AKI condition.

Our platform could achieve the differentiation of healthy, “early AKI” stage and “late AKI” patient by detecting both NGAL and Cystatin C in clinically relevant ranges [10,36,37,38] within 15 min. As NGAL and Cystatin C spike in the serum at different stages of kidney failure [9], our approach combining detection of both these biomarkers within one test allows for injury detection within a large time window, from immediately after the injury to 48 h, reducing the risk of inaccurate diagnosis and improving survival rate in patients experiencing AKI. Our device could perform the detection of analytes within 15 min requiring a serum volume of only 50 µL, which is compatible with its integration into a point-of-care platform available at the intensive care unit and requiring a small amount of blood to perform the analysis [26].

Previous approaches aiming to simultaneously carry out multiple biomarkers’ detection mainly relied on the use of fluorophores embedded in the beads [39,40]. This technology, developed by Luminex, associates a barcode defined as a ratio of fluorophore dye in the bead to each analyte of interest [41], thus allowing for efficient detection of distinct markers in a flow cytometry setup [42]. Despite its performance, the main limitation of this solution is the need to integrate at least two fluorescent filters within the readout platform to ensure barcode reading. This requirement severely hinders a potential integration of this technique in portable devices. Nonetheless, as beads in our DEP-based platform are resolved spatially according to their sizes, the readout can be performed with a single fluorescent channel that could be easily integrated within a PoC device [26]. Another option to optically quantify multiple species without the need for multiple fluorescence channels was proposed by Falconnet et al. [13], who introduced a digitally encoded silicon disk, on the top of which immunoassay would be performed. However, as each barcode has to be observed singularly, the silicon microparticles cannot be accumulated to increase the overall sensitivity of the system. Our method, instead, allows both multimarker analysis and amplification of the outcome signal. Our platform is, thus, the first of its kind, allowing on-chip incubation for optimized analyte collection, multiple biomarkers’ detection, signal amplification, and optical readout by means of a single fluorescence channel.

## 4. Conclusions

In this paper, we presented a fully integrated system for fast and efficient acute kidney injury diagnostics. The introduction of high-aspect-ratio vertical electrodes within the microfluidic channel permitted an accurate manipulation of antibodies-decorated beads through a novel method named “DEP surfing”. The incubation step was conducted in a dedicated area under a high flow rate, ensuring an effective collection of analytes on the surface of beads, an important feature for the detection of analytes at low concentration. We introduced the possibility to perform in the same microchamber, while in two separate phases, sample beads’ incubation and beads’ accumulation, controlled solely by electrical signals. The amplified fluorescent signal acquired with our method proved to be as bright as the one obtained with turbulent mixing. We employed this approach to the concomitant detection of two analytes, thanks to a size-based beads’ separation technique: NGAL and Cystatin C could be simultaneously detected within 15 min in a minimally diluted matrix, and the detection performance matches the one of a commercial ELISA kit. The combined detection of both biomarkers allows for the diagnosis of AKI conditions at different stages, which could be greatly beneficial to patients in intensive care units.

Currently, our device is designed for the detection of two biomarkers. Nonetheless, the technology could be easily readjusted for the detection of more analytes through the use of beads of different sizes. Furthermore, since this system relies on the largely established biochemistry of antibody–beads’ conjugation and on-sandwich assays, it can be easily translated to the analysis of other acute conditions or infectious diseases.

In order to miniaturize the platform and promote its integration in a one-step, point-of-care device, the active fluidic components would need to be replaced by passive fluidic structures, such as capillary pumps, while the beads could be dried in the channels prior to exposure to the patient sample [43,44]. An automated inspection of the cluster size would contribute to the reduction of the inter-experiments’ variability and handling errors.

## 5. Patents

This work resulted in the deposition of the following patent: Thiriet, P.-E.; Medagoda, D.; Guiducci, C. Dielectrophoresis detection device. European Patent (EP) priority: 5 June 2020 no. 20178445.1

## Figures and Tables

**Figure 1 biosensors-10-00212-f001:**
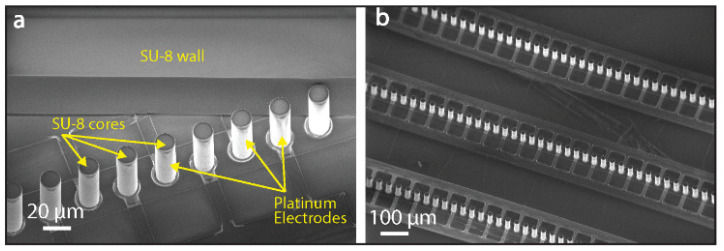
Scanning electron microscope (SEM) images of the concentration lines. (**a**) One of the microbeads’ concentration regions. The beads surf along the diagonal line in the space between the line and the SU-8 photoresist wall and stop at their conjunction. (**b**) Three parallel electrodes’ lines are patterned in the microfluidic channel to allow for accurate deflection of beads driven by the flow. The height of the vertical electrodes and SU-8 microfluidic channels is 50 µm.

**Figure 2 biosensors-10-00212-f002:**
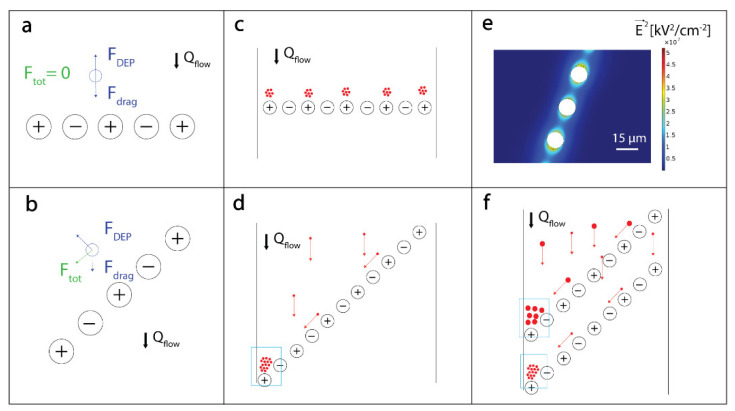
Dielectrophoresis (DEP)-based incubation and accumulation of beads’ working principles and simulation. (**a**,**b**) Illustration of the net force experienced when a bead approaches a horizontal or a diagonal electrodes’ line, respectively resulting in the bead immobilization (**a**) or directed “surfing” along the electric field (**b**). (**c**,**d**) Extending this approach to multiple beads, one can obtain either small clusters for on-chip incubation close to horizontal lines (**c**) or large clusters in the regions where all the surfing beads are immobilized against the microfluidic channel wall (**d**), indicated with a blue rectangle. Beads are depicted in red. (**e**) Finite element simulation of the electrical field generated by electrodes in the diagonal line. The field gradient is higher between electrodes, resulting in a DEP force preventing the beads from crossing the line. Simulations were carried out on Comsol 5.3, using the Electrostatics module and a voltage amplitude difference of 20 V between electrodes. (**f**) Illustration of the multianalyte detection capabilities of our platform. Through the application of different electrical potentials on diagonal lines, we can achieve the clustering of beads of different sizes in separate locations.

**Figure 3 biosensors-10-00212-f003:**
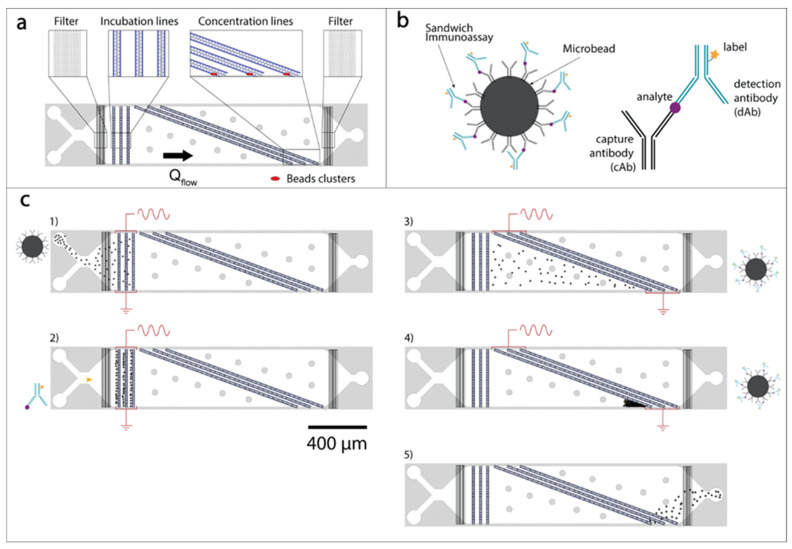
Chip description and operation. (**a**) Presentation of the chip layout. Beads and reagents can be successively injected through the two inlets visible on the left. The device consists of three incubation lines upstream and three concentration lines downstream, at the end of which the beads are accumulated in clusters (shown here in red). (**b**) Illustration of a sandwich immunoassay used for detection of biomarkers. The analyte we aimed to detect was captured by the bead decorated with capture antibody (cAb) and detection was performed thanks to the fluorescently labeled detection antibody (dAb). (**c**) Presentation of the successive steps performed on-chip to operate the platform, namely, (**1**) beads’ loading, (**2**) incubation with detection antibodies and (**3**) release from the incubation line, (**4**) clustering in the concentration region, and (**5**) discarding through the outlet. For the sake of clarity, the species bound to the beads and the electrically activated arrays of electrodes are indicated for each step.

**Figure 4 biosensors-10-00212-f004:**
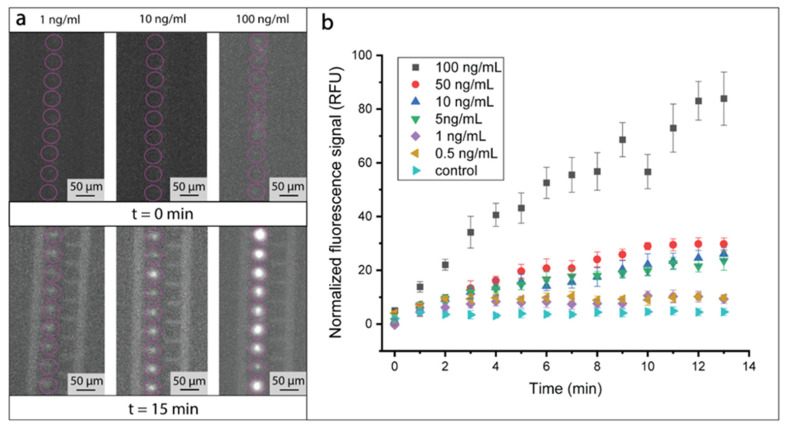
On-chip incubation of Neutrophil Gelatinase-Associated Lipocalin (NGAL) biomarker. (**a**) Observation of the small beads’ clusters (circled in pink) before and after 15 min of incubation. The fluorescence signal arose from the binding of dAb–NGAL complex to cAb-decorated beads dielectrically trapped in the regions upstream to the electrode line. Three NGAL concentrations were injected in separate experiments, namely, 1 ng/mL, 10 ng/mL, and 100 ng/mL. (**b**) Fluorescence signal as a function of the incubation time for different NGAL concentrations. After 15 min all concentrations provided a signal greater than the control experiment, consisting of an injection of a solution in absence of NGAL molecules. The error bars were obtained by measuring the fluorescent signal from 10 clusters.

**Figure 5 biosensors-10-00212-f005:**
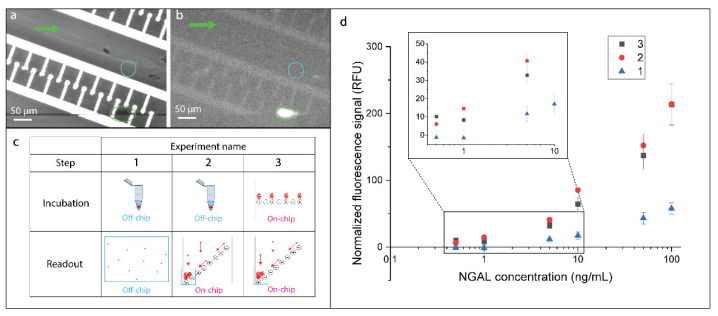
Assessment of the on-chip incubation and accumulation performance. (**a**,**b**) Brightfield (**a**) and fluorescence (**b**) images of beads clustered after 15 min of incubation (NGAL concentration of 100 ng/mL) and concentration steps. A bright fluorescent signal is clearly visible in the accumulation region circled in green. The blue circle indicates the region chosen as neighboring background for the normalization of the cluster signal. (**c**) Experimental protocol implemented to separately assess the impact on the output signal of our platform of both on-chip incubation and accumulation of beads. (**d**) Calibration curves obtained for the aforementioned experiments. The measured relative fluorescence signal is plotted as a function of the NGAL concentration employed for incubation. Plot 3 presents the dose-response of our system with both incubation and accumulation steps performed on-chip and taken as the reference curve in the following discussion section. Error bars were calculated over three acquisitions.

**Figure 6 biosensors-10-00212-f006:**
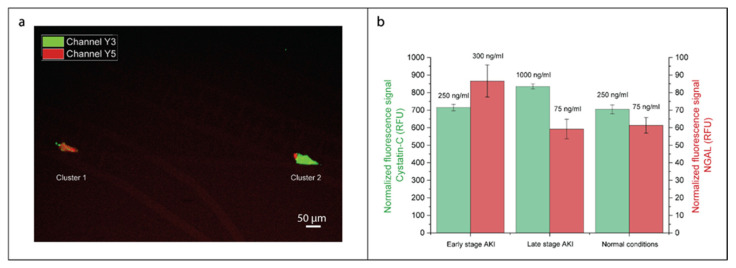
Simultaneous detection of NGAL and Cystatin C for acute kidney injury (AKI) diagnosis. (**a**) Superposition of two clusters acquired, respectively, in CY5 fluorescent channel (NGAL label) and CY3 channel (Cystatin C label). This picture was taken in the case of an “early” patient (NGAL: 300 ng/mL, Cystatin C: 250 ng/mL). (**b**) Fluorescent signal acquired for CY3 channel (Cystatin C detection) and CY5 channel (NGAL detection), respectively, in cluster 2 (downstream, accumulating 2-µm beads capturing Cystatin C) and cluster 1 (upstream, accumulating 6-µm beads capturing NGAL). Three cases were investigated, corresponding to the clinically relevant situations: an “early-stage” patient with high NGAL and normal Cystatin C concentrations, a “late-stage” patient with high Cystatin C and normal NGAL concentrations, and a “healthy” patient with normal NGAL and Cystatin C concentrations. Error bars were calculated over three measurements.

**Table 1 biosensors-10-00212-t001:** Comparison of advantages of our approach with respect to standard methods used for detection of NGAL.

	DEP Surfing	ELISA	Abbott Architect	Lateral Flow Assay
Limit of detection	Low	Low	Low	Average
Total analysis time	Short	Long	Average	Short
Sample processing	Limited	Extensive	Extensive	Limited
Volume needed	Low	Average	Average	High
Multiple analytes	Easy	Difficult	Easy	Average
Translation to PoC	Easy	Difficult	Difficult	Easy

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
