# Peer review of "Rapid Multianalyte Microfluidic Homogeneous Immunoassay on Electrokinetically Driven Beads"

_biosensors, 2020, doi:10.3390/bios10120212_

Round 1
Reviewer 1 Report
The manuscript demonstrates the simultaneous detection of two serum markers (Cystatin C and 46 NGAL) for acute kidney injury (AKI) using the combination of immuno-reaction and dielectrophoresis (DEP) in a microfluidic glass device. Functionalized microbeads were used to capture the analytes which are later labelled with the third fluorescent antibody, forming a conventional sandwich structure. These beads were trapped in the microfluidic device for the capture of the target analytes. Later they were released and aggregated to specific regions near channel side walls. Both trapping and aggregation were achieved by DEP force. The size selective nature of DEP force provides the option of detecting two analytes in a single fluorescent channel. Overall, the reviewer thinks the experiments are well designed. Nevertheless some questions are not answered in the manuscript, preventing its publication in the current form. Specifically,
- How to control the number of beads the were trapped? Understandably, more trapped beads will give higher signal to noise ratio, which can improve the detection of the limit. What number of the beads would be considered “sufficient” and what is the criterion for such determination? Similarly, more uniform control of the bead clusters can offer a more reliable end reading.
- What are the trapping efficiencies for 2 and 6 μm particles? As the authors noticed, some 2 μm particles were mixed with 6 μm particles, while some 6 μm particles were contaminating the 2 μm particles. Based on such observations, there should be some particles escaping the “aggregating process”. How does such uncertainty affect the interpretation of the final analyte concentrations?
- Please quantify the contamination of the beads, for example red in green and green in red. Figure S5 provides some insight. Quantification the ratio of the cross-contamination can help. Also, more detailed discussion of how such contamination can affect the terminal outcome is required.
- The reviewer did not see the limit of detection of this method. As this method working as a integrated sensor, this critical parameter should be provided and discussed in the text.
- Why multiple flow rates (2 μL/min, 0.4 μL/min and 0.8 μL/min) were used and how were these flow rate chosen?
- Disposable or reusable?
- How were the fluorescent signals normalized?
- Figure S4/S6, please provide calibration curves for both analytes in question. The reviewer feels the plots of calibration curves should be in the main text as it is of the direct interests for readers who want to know its applicability in the real-world use.
- How were the fluorescent signals normalized?
- Please explain what the error bars are in each relevant figure.
- Authors may provide a table comparing their approach with existing ones to show their advantages and limitations.
- Introduction seems a bit too lengthy and not concentrated.
Author Response
Dear reviewer,
Please see the attachment.
Best regards,
Pierre-Emmanuel Thiriet

Reviewer 2 Report
This work on the development of an homogeneous immunoassay for two acute kidney injury (AKI) biomarkers with the application of dielectrophoretic (DEP) forces to drive decorated beads in a microfluidic chip, is extremely interesting and well presented. Analytical data are convincingly supporting the conclusions. The microfluidic chip is well characterized and comparison among different methods already published has been included.
Please, find minor comments here below:
-NGAL should be written in full both in the abstract and in the text.
-the protocol for the immunoassay which is illustrated in Figure 3, could be better understood if the corresponding assay steps are reported in the text as a list with different steps well separated and explained.
Author Response

(The authors gave the same response as above.)

Round 2
Reviewer 1 Report
The authors have addressed most of my previous concerns. Some outstanding ones are as follows,
- Is it practical to visually assess the cluster size every time? What is the uniformity of the cluster sizes for both 6 and 2 μm particles? Are there any other ways to control the cluster size more objectively,reliable, and free of the human errors?
- The reviewer does not see the trapping efficiencies for the two particles. If 100 particles are injected into the chip, how many of them would be trapped in the clusters? This might not be critical as particles are not costly. But the from the technical point of view, this is interesting.
- Authors may include the information that the device can be used 20 times.
- Can the authors indicate the region used as negative control?
- In Figure 4a 100 ng/mL, there is a spot without cluster. What is the reason behind this missing cluster and how frequent this can occur?
- Authors clarified that different flow rates were used for different particles. From line 225-226, it seems particle mixture of the two sizes was injected into the device. In this case, did the authors change the flow rate during the trapping process?
Author Response

(The authors gave the same response as above.)
